# A time-resolved multi-omics atlas of *Acanthamoeba castellanii* encystment

Clément Bernard[1,7], Marie Locard-Paulet [2,7], Cyril Noël [3,7], Magalie Duchateau [4], Quentin Giai Gianetto[4,5], Bouziane Moumen[1], Thomas Rattei [6], Yann Hechard[1], Lars Juhl Jensen [2], Mariette Matondo [4] & Ascel Samba-Louaka [1✉]

Encystment is a common stress response of most protists, including free-living amoebae. Cyst formation protects the amoebae from eradication and can increase virulence of the bacteria they harbor. Here, we mapped the global molecular changes that occur in the facultatively pathogenic amoeba *Acanthamoeba castellanii* during the early steps of the poorly understood process of encystment. By performing transcriptomic, proteomic, and phospho-proteomic experiments during encystment, we identified more than 150,000 previously undescribed transcripts and thousands of protein sequences absent from the reference genome. These results provide molecular details to the regulation of expected biological processes, such as cell proliferation shutdown, and reveal new insights such as a rapid phospho-regulation of sites involved in cytoskeleton remodeling and translation regulation. This work constitutes the first time-resolved molecular atlas of an encysting organism and a useful resource for further investigation of amoebae encystment to allow for a better control of pathogenic amoebae.

[1] Laboratoire Ecologie et Biologie des Interactions, Université de Poitiers, UMR CNRS, 7267 Poitiers, France. [2] Novo Nordisk Foundation Center for Protein Research, University of Copenhagen, Copenhagen, Denmark. [3] IFREMER-IRSI-Service de Bioinformatique (SeBiMER), Centre Bretagne, Plouzane, France. [4] Institut Pasteur, Université de Paris, Proteomics Platform, Mass Spectrometry for Biology Unit, UAR2024, CNRS 2000 Paris, France. [5] Institut Pasteur, Université de Paris, Department of Computation Biology, Bioinformatics and Biostatistics Hub, Paris, France. [6] Centre for Microbiology and Environmental Systems Science; Doctoral School Microbiology and Environmental Science, University of Vienna, Vienna, Austria. [7] These authors contributed equally: Clément Bernard, Marie Locard-Paulet, Cyril Noël. ✉email: ascel.samba@univ-poitiers.fr

Encystment is one of the most prevalent differentiation processes in the living world[1]. This response to environmental stress allows many unicellular organisms to transform into a dormant and protected form resistant to damage called a cyst. Among protists, most free-living and parasitic amoebae, such as *Acanthamoeba castellanii* and *Entamoeba histolytica*, can undergo encystment[1]. This ensures species sustainability by allowing them to survive in the environment due to the high resilience of the cysts[1].

FLA are protists commonly found in water and soil[2]. They are unicellular, chemoheterotrophic, and phagotrophic organisms. Some are amphizoic as they can live and thrive in the environment but can also invade humans. Some FLA cause opportunistic and non-opportunistic infections. For example, the genus *Acanthamoeba* was reported to cause fatal amoebic meningoencephalitis[3]. Several *Acanthamoeba* species were also reported to cause a painful corneal disease named *Acanthamoeba* keratitis[4]. This disease typically occurs in contact lens wearers, linked to poor contact lens hygiene, but also after trauma and contact with contaminated water[5].

Beyond the pathogenesis they directly induce, FLA are considered reservoirs of pathogenic bacteria. Indeed, FLA feed on bacteria by phagocytosis[2]. These bacteria are mainly digested, but some of them, collectively named amoeba-resisting bacteria, can survive grazing by FLA[6]. Bacteria that are retained within FLA cysts become *de facto* protected from biocides[6]. In addition, some bacteria have been shown to become more virulent and more resistant to biocides and antibiotics after their passage through amoebae[6].

Although FLA are considered a public health concern, our knowledge regarding their encystment is currently limited. FLA life cycle can exhibit two forms: trophozoite and cyst. The trophozoite is the vegetative and biochemically active form that feeds and divides. Under adverse environmental conditions or stress, FLA differentiate from trophozoite to cyst, the resting form. This process named encystment is reversible, and FLA may excyst under favorable environmental conditions. The cyst of some FLA, such as *Acanthamoeba*, is highly resistant to various antimicrobial agents, disinfection agents, desiccation, and radiation[7].

Encystment is characterized by morphological and biochemical modifications with a reduction of metabolic activities. Proteins involved in signal transduction such as the protein kinase C are highly expressed and a rise in cyclic AMP (cAMP) level is associated with a high activity of adenylate cyclase and phosphodiesterase[8–10]. At the biochemical level, sugar metabolism is important, since glycogen is rapidly degraded for cyst-wall synthesis[11]. Thus, glycogen phosphorylase is required for cyst-wall assembly[12]. In addition, cellulose synthase and xylose isomerase have also been shown to play a role in cyst-wall formation[13]. Several proteases have been reported to promote encystment along with activation of autophagy to allow for rapid proteome remodeling[14–17]. With a reduction in DNA synthesis and RNA levels by about 50% within the cyst[18] encystment induces a reduction in cell metabolism that is associated with a reduction of cytoplasmic volume leading to the retraction of the cytoplasm from the cyst wall[11].

Although transcriptomic and proteomic analyses have been conducted on free-living amoebae[17,19], understanding of the signaling pathways involved in their stress response remains scarce and fragmented. Only a few annotated genomes are available for amoebae. Added to that, there is also limited knowledge of their transcriptomic landscape since their annotations do not span different environmental conditions, or biological context. For these reasons, we investigated in this study the encystment of *A. castellanii*, a model eukaryote for cellular biology, as it readily undergoes encystment in starvation conditions

and has an annotated genome available[5]. We developed a proteogenomics approach[20] that combines transcript- and protein-level analyses, to identify protein sequences beyond the ones already reported in the proteome of reference[21,22] and monitor the early steps of *A. castellanii* encystment in vitro.

We first performed RNA-seq to identify genes expressed in the specific context of encystment, after one, four, and eight hours of starvation. This led to the identification of more than 150,000 previously undescribed transcripts that were used together with available protein-level annotations to analyze mass spectrometry (MS)-based proteomics data. To understand the cellular signaling events that govern encystment, we also analyzed the phosphorylation site regulations occurring in the same conditions by mass spectrometry. This resulted in a comprehensive molecular atlas of the early steps of *A. castellanii* encystment. These transcriptomic, proteomic and phosphoproteomic data sets provide a valuable overview of the regulation events that occur during the early steps of FLA encystment. We believe that the community can build upon this work to improve proteome annotation of *A. castellanii* and find molecular drivers of encystment.

## Results

**Transcriptomic landscape of *A. castellanii* encystment in vitro.** As one of the reference models for studying encystment, *A. castellanii* strain Neff was used for this study. We validated its capacity to encyst in a nutrient-devoid Tris-based medium described by Neff et al.[23]. This approach was originally designed to induce the synchronous encystment of *Acanthamoeba* by starving them while maintaining a constant pH, between 8.6 and 9.0. We followed the apparition of cysts by flow cytometry using calcofluor white (Supplementary Fig. 1a, b), which binds to $\beta-1,3$ and $\beta-1,4$ polysaccharides, including cellulose, one of the main components of *A. castellanii* cyst cell-wall. Cysts were observed as early as 24 h after encystment induction ($\approx$20% of the cell population). After a week, 87% of the cell population was encysted (Supplementary Fig. 1c).

The approach designed for multi-OMICs analysis of encystment at the transcriptome, proteome, and phosphorylation level is presented in Fig. 1 (see the "Methods" section for a more detailed description). We chose to focus on signaling pathways and molecular mechanisms involved in the early steps of the encystment process. Thus, RNA and proteins were harvested 1, 4, and 8 h after triggering encystment. We considered the 1 h time-point because it precedes the maximum expression of the trehalose synthase gene observed at 2 h in encysting cells[24]. We opted for 4 and 8 h, as Hirukawa et al. did not detect the cyst-specific protein CSP21, a marker of *A. castellanii* encystment, at these time points[25]. The control condition (T0 set as the time before switching to encysting medium) was used as a reference throughout the analysis. Control cells were grown in nutrient-complete growth medium (PYG), in contrast to the nutrient-devoid starvation tris-based medium used to induce encystment.

We estimated the reference genome completeness to be 77% (Table 1). Thus, to increase coverage and identify new transcripts, we generated a transcriptome de novo for which we obtained a completeness score of 94% (Table 1). This transcriptome contained 166,782 transcripts for 55,723 annotated genes (Table 2). This number is much higher than the 15,455 genes already reported[5]. Such a difference could result from an overestimation of the number of transcripts assembled by Trinity due to non-coding RNAs or isoforms. We used the 166,782 identified transcripts to perform a InterProScan analysis that resulted in 121,821 matches, whereas the BLASTx (non-redundant protein database) search yielded 145,978 hits (Fig. 1, green pipeline). The InterProScan analysis of our transcriptome

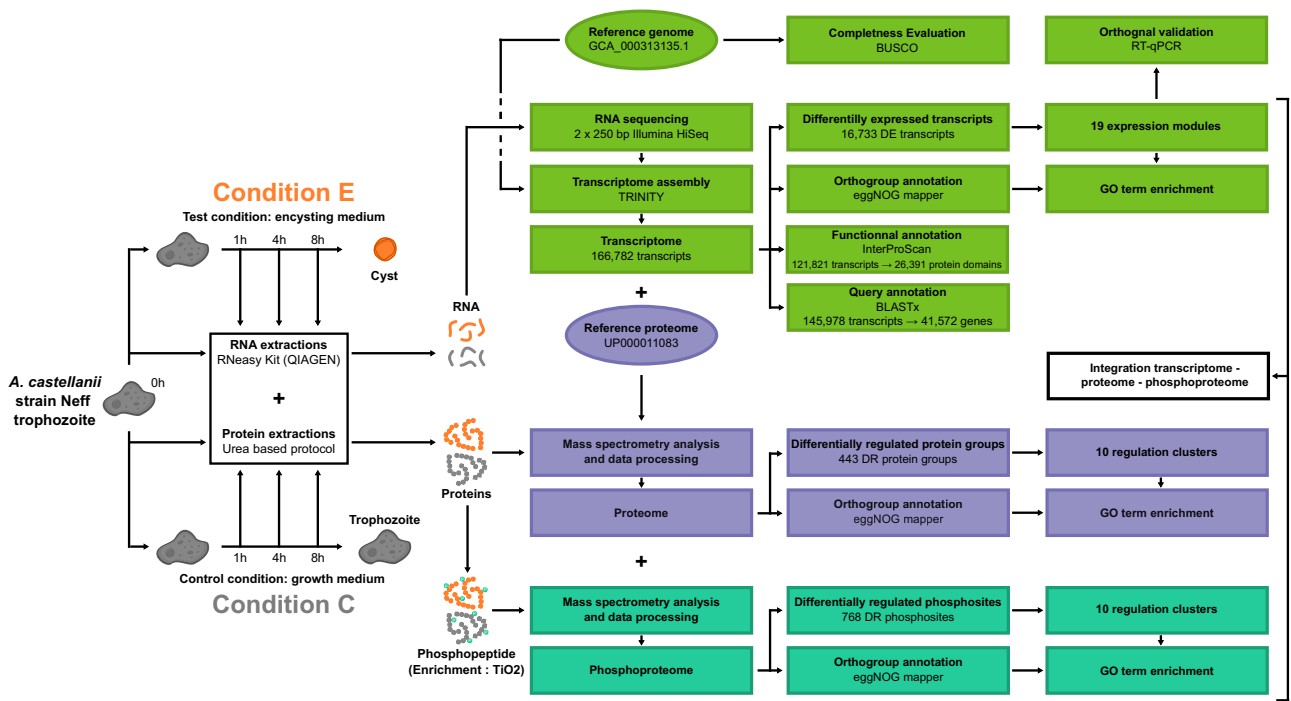

**Fig. 1 Multi-OMICs characterization of *A. castellanii*.** Experimental design utilized for studying *A. castellanii* encystment in vitro. Samples were harvested at 0 h (reference sample — T0), 1, 4 and 8 h after triggering encystment (E) as well as in the control condition (C). RNA and proteins were extracted. Proteins were trypsin-digested, and phospho-peptides were enriched. The three samples were analyzed with the pipeline described on the right (see material and methods for more detailed information). DR differentially regulated, DE differentially expressed, GO gene ontology.

suggested the presence of 26,391 protein domains against 41,572 genes using BLASTx. Mapping these outputs to the mass spectrometry data described in the next sections allowed us to focus on detectable proteins in the next parts of this manuscript.

We explored the relative transcript abundances during encystment. DESeq2 identified 16,733 differentially expressed transcripts (false discovery rate ≤ 0.001; 8542 genes according to the Trinity annotation)[26]. These were clustered using Weighted Correlation Network Analysis (WGCNA)[27], which resulted in 19 expression modules containing transcripts presenting similar regulation patterns during encystment (Supplementary Data 1). Figure 2a and Supplementary Fig. 2 present the mean log2-transformed fold change relative to T0 and independent transcript regulations, respectively. Four main modules gathered more than 95% of the regulated transcripts. In modules 1 and 2, transcripts were upregulated as early as 1 h after treatment and remained up during encystment, in comparison to no change in the control condition. Modules 18 and 19 (more than 50% of the differentially expressed transcripts) presented the opposite trend: transcripts were downregulated as early as 1 h after triggering encystment and remained down in module 18 or returned to baseline in module 19, relative to control. The other modules displayed more dynamic time-dependent patterns. For example, transcripts in module 9 were downregulated at 1 h but returned to the baseline in the growth condition and were upregulated 1 h before being under-expressed at 4 and 8 h during encystment.

We randomly selected 25 transcripts from 16 of the expression modules for orthogonal validation using RT-qPCR. Primer pairs were designed to target transcripts with only one isoform considered as differentially expressed. RT-qPCR was performed on RNA extracts obtained using the same protocol as that for RNA-seq analysis. Two genes already known to be involved in encystment were used as controls: CSP21 and trehalose synthase[24,25]. CSP21 mRNA has been shown to increase gradually, rise between 12 and 20 h after encystment induction,

and decrease when cysts reach maturity[25]. For trehalose synthase, gene expression rises swiftly after induction, and Bínová et al. observed a maximum expression at 2 h, before a return to baseline after 12 h[24].

80% of the expression profiles from the genes examined by RT-qPCR displayed comparable trends to that obtained by RNA-seq (Supplementary Fig. 3). This indicates a good reproducibility between these orthogonal methods. Several discrepancies were expected, as differences in gene expression profiles obtained with RNA-seq and RT-qPCR techniques have been reported as a general phenomenon[28,29]. We further resolved these discrepancies by comparing the RNA-seq data against protein-level relative quantities in the same experimental conditions (see Proteomic section below).

As functional annotation remains very sparse for *A. castellanii*, we used eggNOG-mapper[30,31] to transfer functional information from orthologous sequences to unannotated transcripts. With these, we performed Gene Ontology (GO) term enrichment for each module to highlight the biological processes at play during encystment (see Supplementary Data 2 for enrichment results, Supplementary Data 3 file for the functional annotation of the transcripts and the material and methods for more information on the statistical approach). The GO terms enriched, according to our statistical thresholds, are grouped according to their functions in Fig. 2b (only biological processes are presented for clarity). This analysis returned significant enrichments in modules that contain upregulated (modules 1, 2, 13, 16) as well as down-regulated (modules 18, 19) transcripts during encystment.

Unsurprisingly, many processes related to cell proliferation were enriched in the downregulated modules: module 19 was enriched for transcripts involved in mitosis, replication, DNA repair, and DNA metabolic process. Module 18 was enriched for genes that could explain the metabolic shutdown during encystment. GO terms related to RNA metabolic processes were enriched in this module, as well as ribosome biogenesis, and

**Table 1 Completeness analysis of *A. castellanii* transcriptome against published genomes of *A. castellanii* and close orthologs using BUSCO.**

| Reference genome vs BUSCO eukaryota_odb10 database (255 single-copy genes) | Complete | Complete and single-copy | Complete and duplicated | Fragmented | Missing |
|---|---|---|---|---|---|
| *Acanthamoeba castellanii* strain Neff [GCA_000313135.1] | 77.3% (197) | 76.5% (195) | 0.8% (2) | 5.1% (13) | 17.6% (45) |
| *Acanthamoeba castellanii* strain Namur [GCA_903821525.1] | 78.9% (201) | 72.2% (184) | 6.7% (17) | 11.4% (29) | 9.7% (25) |
| *Acanthamoeba polyphaga* strain Linc Ap-1 [GCA_001567625.1] | 61.2% (156) | 60.0% (153) | 1.2% (3) | 22% (56) | 16.8% (43) |
| *Dictyostelium discoideum* [GCA_000004695.1] | 92.2% (235) | 91.8% (234) | 0.4% (1) | 2.4% (6) | 5.4% (14) |
| *Naegleria gruberi* [GCA_000004985.1] | 79.6% (203) | 79.2% (202) | 0.4% (1) | 5.5% (14) | 14.9% (38) |
| **Our transcriptome vs BUSCO eukaryota_odb10 database (255 Single-copy genes)** | **Complete** | **Complete and Single-copy** | **Complete and duplicated** | **Fragmented** | **Missing** |
| *Acanthamoeba castellanii* strain Neff [This study] | 94.2% (240) | 42.0% (107) | 52.2% (133) | 1.6% (4) | 4.2% (11) |

The completeness to a set of single-copy protist orthologs was assessed using BUSCO, version 3.0.2[61].

**Table 2 Comparison of the coverage of the transcriptome generated in this study and the reference transcriptome.**

| Data | Nb. of transcripts | Nb. of genes | Lenght (bp) | Max (kbp) | N50 | GC% | Mean cov. |
|---|---|---|---|---|---|---|---|
| Transcriptome *Acanthamoeba castellanii* strain Neff (This study) | 166,782 | 55,723 | 388,816,638 | 45,204 | 4260 | 60.61 | 271X |
| Transcriptome *Acanthamoeba castellanii* strain Neff (Clarke et al.[5]) | 14,981 | 15,655 | 19,742,193 | 13,440 | 1722 | 62.94 | |

Some metrics were calculated from the reference genome using Assemblathon v2 (https://github.com/KorfLab/Assemblathon/).

many different cellular metabolic processes related to a large panel of different molecules (ribonucleotide, sulfur compound, amide, amino acids, etc.).

Among modules containing transcripts upregulated during encystment (1, 2, 13, and 16), both 2 and 13 were enriched for GO terms related to carbohydrate processes (GO:005975—carbohydrate metabolic process, GO:0034637—cellular carbohydrate biosynthetic process). This may reflect the formation of the amoeba cellulose cell wall during encystment.

Several GO terms related to the proteasome were enriched in modules 1 and 16. Both ubiquitin-dependent and independent proteasome-mediated processes were enriched in these modules (GO:0010499—proteasomal ubiquitin-independent protein catabolic process, GO:0032436—positive regulation of proteasomal ubiquitin-dependent protein catabolic process, GO:0043248—proteasome assembly, GO:0043161—proteasome-mediated ubiquitin-dependent protein catabolic process). This confirms the importance of proteasomal degradation during encystment.

**Proteome remodeling during *A. castellanii* cyst formation.** We performed proteomic analysis of *A. castellanii* under the same culture conditions that we used for the transcriptome. To match MS spectra to peptide sequences, we used the proteome of reference of *A. castellanii* strain Neff[5], which we completed with protein sequences identified from the transcriptome (Fig. 3a). With this custom database we identified 8577 protein groups (proteins that cannot be distinguished by MS because identified by a common set of peptides) (Fig. 3a) among which 2342 had no functional annotation retrieved with eggNOG-mapper. 5129 protein groups were present in both our transcriptome and the proteome of reference, 747 were only identified in the reference proteome, and 2701 were specifically identified from sequences of the transcriptome of *A. castellanii* encystment. This highlights the importance in combining context-specific transcriptomic data to analyze the proteome of poorly annotated organisms.

Proteins were analyzed together with their most intense transcript to get a combined view of RNA- and protein-level regulation. A total of 3659 protein groups were quantified in both the proteome and the transcriptome with a minimum number of successive measurements ≥3 in a minimum of two replicates in the MS data. Among these, 827 were regulated only at the transcript level, 443 were specifically regulated in the proteome (ANOVA $q$-value ≤ 0.05 and minimum one time point with a two-sided Welch paired $t$-test corrected $p$-value ≤ 0.05), and 133 were regulated across encystment in both transcriptomic and proteomic data sets (Fig. 3a). Figure 3b shows the distribution of the log2-transformed fold changes relative to T0 observed for the protein groups and the corresponding transcripts in purple and green, respectively. In contrast to the changes at the transcriptome level, downregulation at the proteome level was delayed and occurred gradually across the three time points with a peak at the eight-hour timepoint (Fig. 3b). These observations were corroborated by the principal component analysis (PCA) performed on the transcriptomic and proteomic data (Supplementary Fig. 5), and the volcano plots of the proteomic analysis (Supplementary Fig. 6). The transcriptome and proteome data corresponding to the 60 protein groups the most regulated in both data sets are presented in Supplementary Fig. 4d. Except for a few protein groups, their transcriptome regulation occurred before 1 h in the encystment condition, resulting in protein-level changes at later time points.

We identified a total of 443 protein groups significantly regulated during cyst formation, which were clustered according to their relative quantities across the three time points in the control and encystment condition (Fig. 3c, see material and methods for more detailed analysis and Supplementary Data 4). As previously mentioned, few proteins showed regulation in the control condition, except in the clusters 4, 9, and 10, which contained altogether 48 protein groups. During encystment cluster 2 contained the highest number of downregulated protein

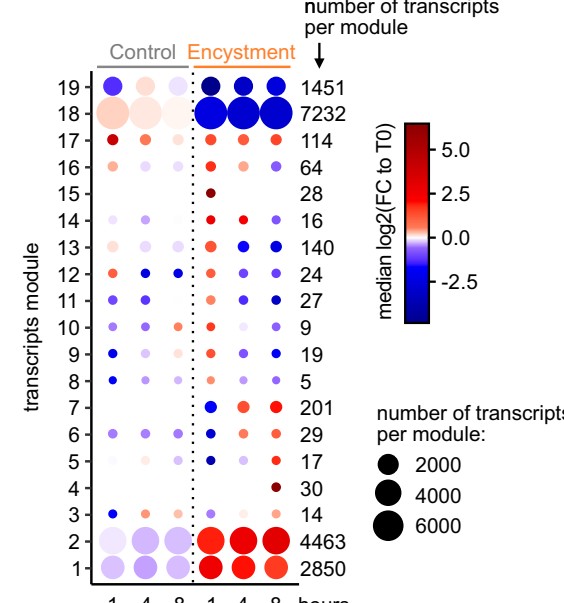

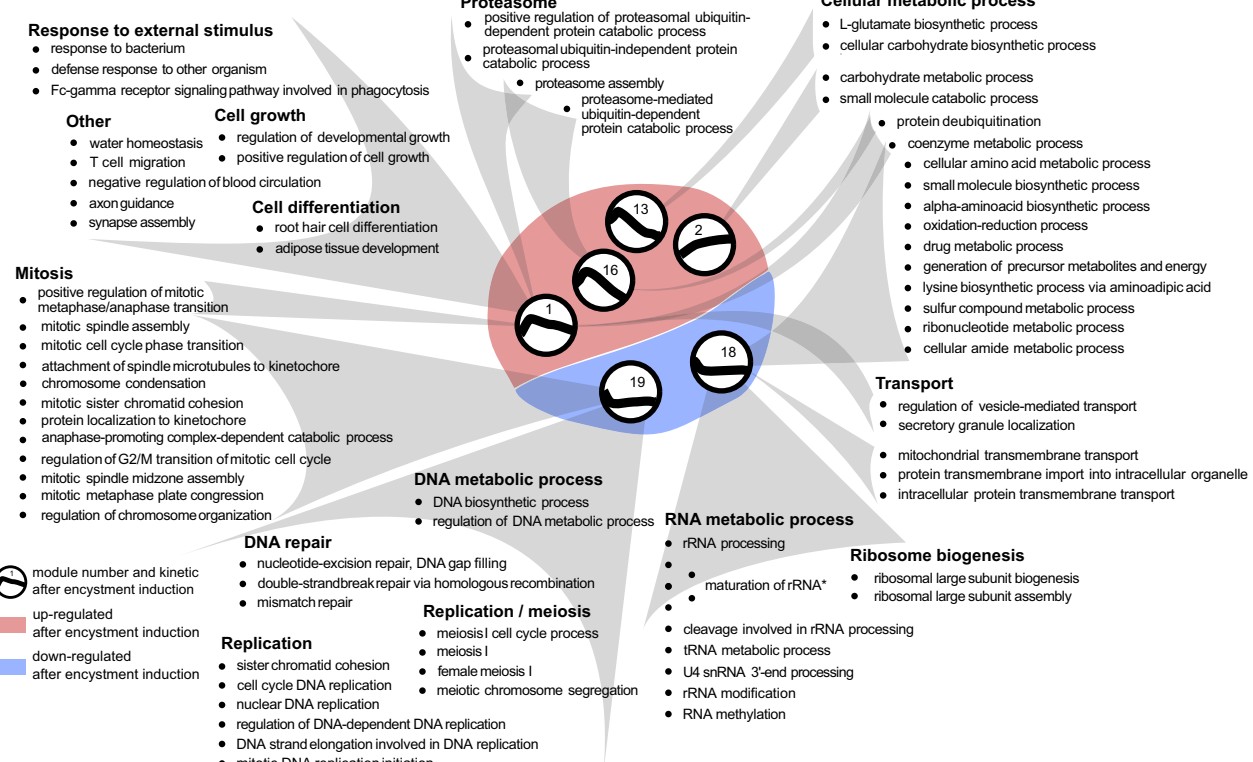

**Fig. 2 Transcriptomic landscape of *A. castellanii* encystment in vitro. a** Modules of the differentially expressed transcripts. The dot plot is color-coded with the mean of log2-transformed fold change relative to the control condition (T0) for each module during encystment and in the control condition (1, 4, 8 h). The size of each dot is proportional to the number of transcripts per cluster (indicated on the right). **b** Functional annotations retrieved with eggNOG-mapper and significantly enriched in the given expression modules. The module median kinetics in the encystment condition are represented in the middle (upregulation and downregulation in the red and blue area, respectively). The detailed outputs of the enrichment can be found in Supplementary Data 2 file. "*": maturation of 5.8S rRNA; maturation of SSU-rRNA from tricistronic rRNA transcript (SSU-rRNA, 5.8S rRNA, LSU-rRNA); maturation of LSU-rRNA from tricistronic rRNA transcript (SSU-rRNA, 5.8S rRNA, LSU-rRNA); maturation of 5.8S rRNA from tricistronic rRNA transcript (SSU-rRNA, 5.8S rRNA, LSU-rRNA); endonucleolytic cleavage in ITS1 to separate SSU-rRNA from 5.8S rRNA and LSU-rRNA from tricistronic rRNA transcript (SSU-rRNA, 5.8S rRNA, LSU-rRNA).

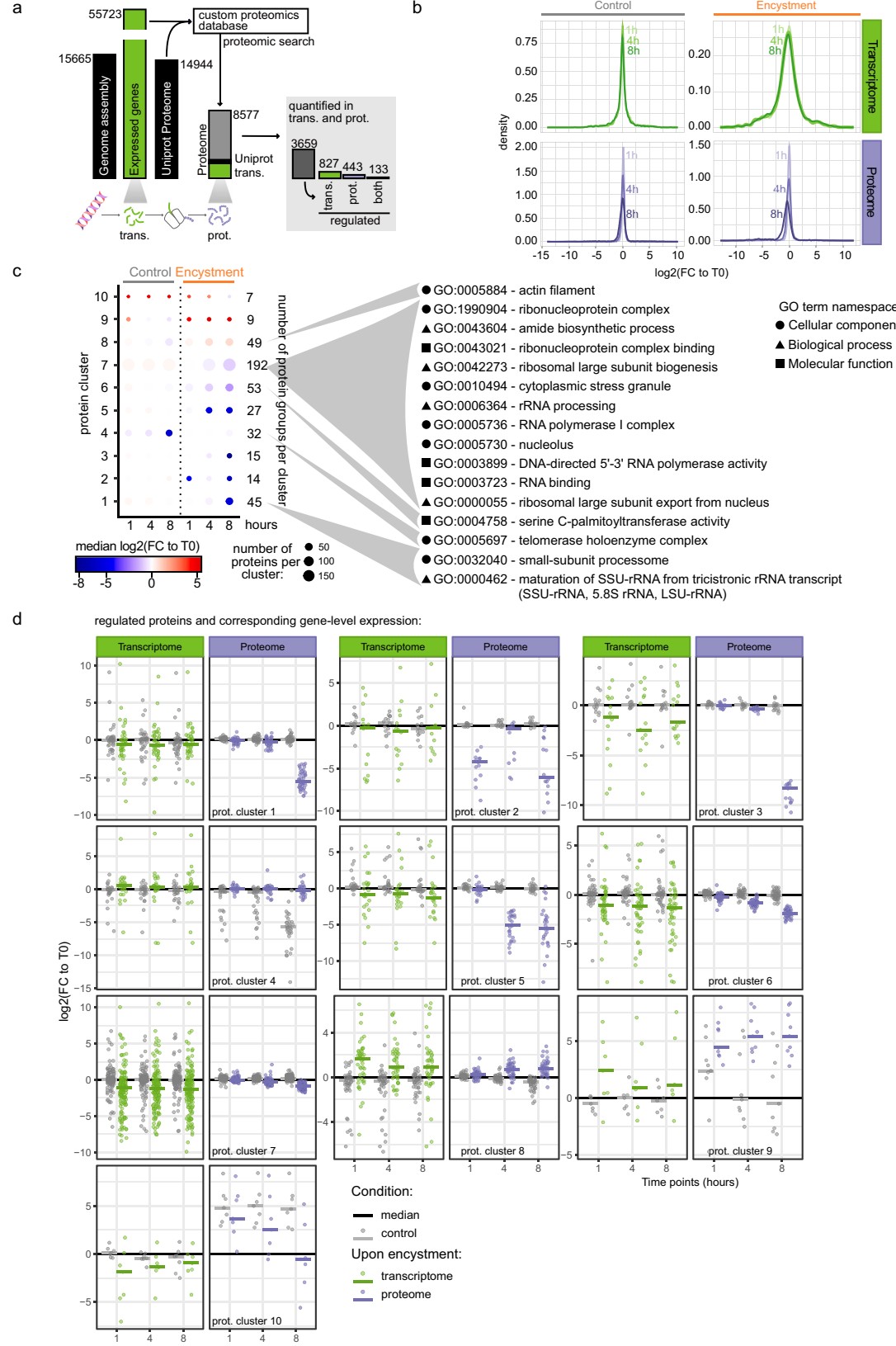

groups at 1 h, whereas other clusters such as cluster 1 and 5 showed downregulation at later time points. Clusters 8, 9, and 10 gather all the protein groups upregulated during encystment (65 protein groups) Fig. 3c.

We performed a GO term enrichment analysis of the proteomic clusters (Supplementary Data 2 and Fig. 3c), which

returned only one significant enrichment for upregulated clusters: actin filament (GO:0005575) in cluster 8. This cluster gathers protein groups that were mainly regulated after encystment induction, with minor changes of protein quantities in the control condition. The presence of this GO term is consistent with the heavy changes in the cytoskeleton organization known to occur

**Fig. 3 Proteome remodeling during *A. castellanii* encystment. a** Proteogenomic pipeline and proteome coverage. The number of genes identified in the genome of reference of *A. castellanii* strain Neff as well as in the de novo transcriptome are plotted next to the number of protein sequences in the proteome of reference in Uniprot. These were utilized to generate a custom proteomics database for protein identification by MS/MS. In this data set, we identified 8677 protein groups matched to sequences from the proteome of reference ("Uniprot"), the transcriptome ("trans.") or present in both (gray). The protein groups quantified at both transcript and protein ("prot.") level are presented on the right, next to the number of protein groups significantly regulated in the transcriptome (green), le proteome (purple) or both (black). **b** Normalized distribution of the log2-transformed fold changes relative to T0 control in the proteome (purple) and the corresponding transcripts (green) in the control condition (left) and during encystment (right). Time points are indicated next to the corresponding density. **c** Clusters of significantly regulated protein groups. The dot plot is color-coded with the mean of log2-transformed fold change relative to the control condition (0 h) for each cluster during encystment and in the control condition (1, 4, 8 h). The size of each dot is proportional to the number of protein groups per cluster (indicated on the right). Significantly enriched GO terms in the different protein clusters are indicated on the right side of the plot (cellular component, biological process and molecular function are represented with a circle, a triangle and a square, respectively). The detailed outputs of the enrichment can be found in Supplementary Data 2 file. **d** Comparison of transcript- and protein-level regulation. The log2-transformed fold changes relative to T0 of protein groups from each cluster presented in (**c**) were plotted next to the corresponding transcripts at the three time points (*x*-axis) in the control (gray) and encystment condition (transcripts and protein groups are in green and purple, respectively). Median values are presented by a horizontal bar.

during cyst formation. For downregulated protein groups during encystment, GO term enrichment results were similar to the ones in the transcriptome. Processes involving rRNA processing (GO:0006364), maturation of SSU-rRNA from tricistronic rRNA transcript (GO:0000462), RNA polymerase I complex (GO:0005736) and RNA binding (GO:0003723), or GO terms associated to ribosome biogenesis (GO:0042273/GO:0000055) were downregulated. This could also be associated with the *A. castellanii* metabolic shutdown that is known to occur during encystment.

Surprisingly, when we matched the regulated protein groups to their most intense transcript, only a limited number of protein-level changes corresponded to a significant regulation at the transcript level (only 22% of the regulated protein groups were matched to significantly regulated transcripts). It is possible that some transient expression changes occurred before 1 h or between our time points and would be missed in the transcriptome. Figure 3d shows protein- and corresponding transcript-level relative quantities in the different clusters. Most of the regulated protein groups are in clusters 6 and 7, which present similar kinetics across encystment: downregulation of the transcripts as soon as 1 h after triggering encystment, and a slower decrease of the corresponding protein quantities peaking at 8 h.

**Phosphorylation-level regulations underlying *A. castellanii* cyst formation.** Phosphorylation is one of the most common post-translational modifications in eukaryotic cells[32]. It is involved in numerous cellular processes, so we sought to investigate its importance in *A. castellanii* cyst formation. We enriched for phosphopeptides using titanium dioxide ($TiO_2$) and analyzed them using tandem MS alongside the non-enriched proteome[33,34]. In the phosphoproteome, 6376 phosphorylation sites were identified (Supplementary Data 5). The distribution of the signal of the corresponding protein groups in the proteome shows that these were not over-represented in the most abundant proteins, indicating that our phospho-enrichment strategy was not biased by protein abundance and that we were able to capture phosphopeptides from low-abundance proteins (Fig. 4a).

The relative quantities of phosphopeptides depend on phospho- as well as protein-level regulation. Since we showed that several hundred proteins are downregulated during encystment, it was perhaps not surprising to also observe global downregulation of phosphopeptide signal during encystment (Fig. 4b). However, many of these were downregulated as soon as one hour after encystment induction in contrast to the slower kinetics of protein downregulation observed in our proteomics

analysis. This suggests that phosphopeptide downregulation results from active dephosphorylation (Fig. 4b). To reduce the bias due to protein-level regulation in the analysis of the phosphoproteome, we performed ANOVA contrast analyses between the relative quantities of protein and phosphopeptide and excluded the phosphorylation sites that presented a significantly similar regulation pattern at protein- and phosphopeptide-level (see material and methods for more details). Hence, we only considered phosphorylation sites regulated across encystment if they were (1) significantly regulated over the time course between encystment and control ($q$-value ≤ 0.05 and minimum of one two-sided paired $t$-test with a $q$-value ≤ 0.05 between the two conditions at each time point), and (2) were significantly different from the time course of the protein group they belong to ($q$-value ≤ 0.05 in the two conditions). According to these thresholds, 768 phosphosites were significantly regulated (up or down) during encystment in these data.

We clustered the regulated phosphosites according to their relative abundance across encystment (Fig. 4c). Of the 10 clusters, clusters 2, 5, 9, and 10 were the only ones to display pronounced changes in the control condition. Unlike clusters 9 and 10, which contain phosphosites that behave similarly in control and encysting conditions, clusters 2 and 5 presented different regulation patterns between control and encystment. Clusters 3, 4, 6, 7, and 8 contain 414 phosphosites downregulated upon encystment, and clusters 1, 2, 5, 9 and 10 contain 285 phosphosites that significantly increased after encystment induction. This confirmed the observations made in Fig. 4b that there was a noticeable trend toward a decrease in phosphorylation site abundance over time.

This could be explained by an increase in phosphatase activity. To test this hypothesis, we conducted a phosphatase activity assay on protein samples obtained either 1 h after encystment induction, or 1 h after growth medium renewal as a control (Fig. 4d). Results show an increased phosphatase activity after encystment induction (0.5 pmol phosphate/min/μg protein) compared to the control condition (0.23 pmol phosphate/min/μg protein) (Mann–Whitney test: *p*-value = 0.0325). These results suggest that an active phosphatase-driven dephosphorylation of specific proteins takes place from initiation during *A. castellanii* encystment.

The GO term enrichment analysis carried out on the phosphoproteome clusters is available in the Supplementary Data 2 file and summarized in Fig. 4c. Some GO terms associated with cytoskeleton remodeling (GO:0051015— actin filament binding, GO:0030864— cortical actin cytoskeleton, GO:0120105—actomyosin contractile ring, intermediate layer) were enriched in clusters 3 and 9, which display a decrease and an increase of phosphosites abundance upon

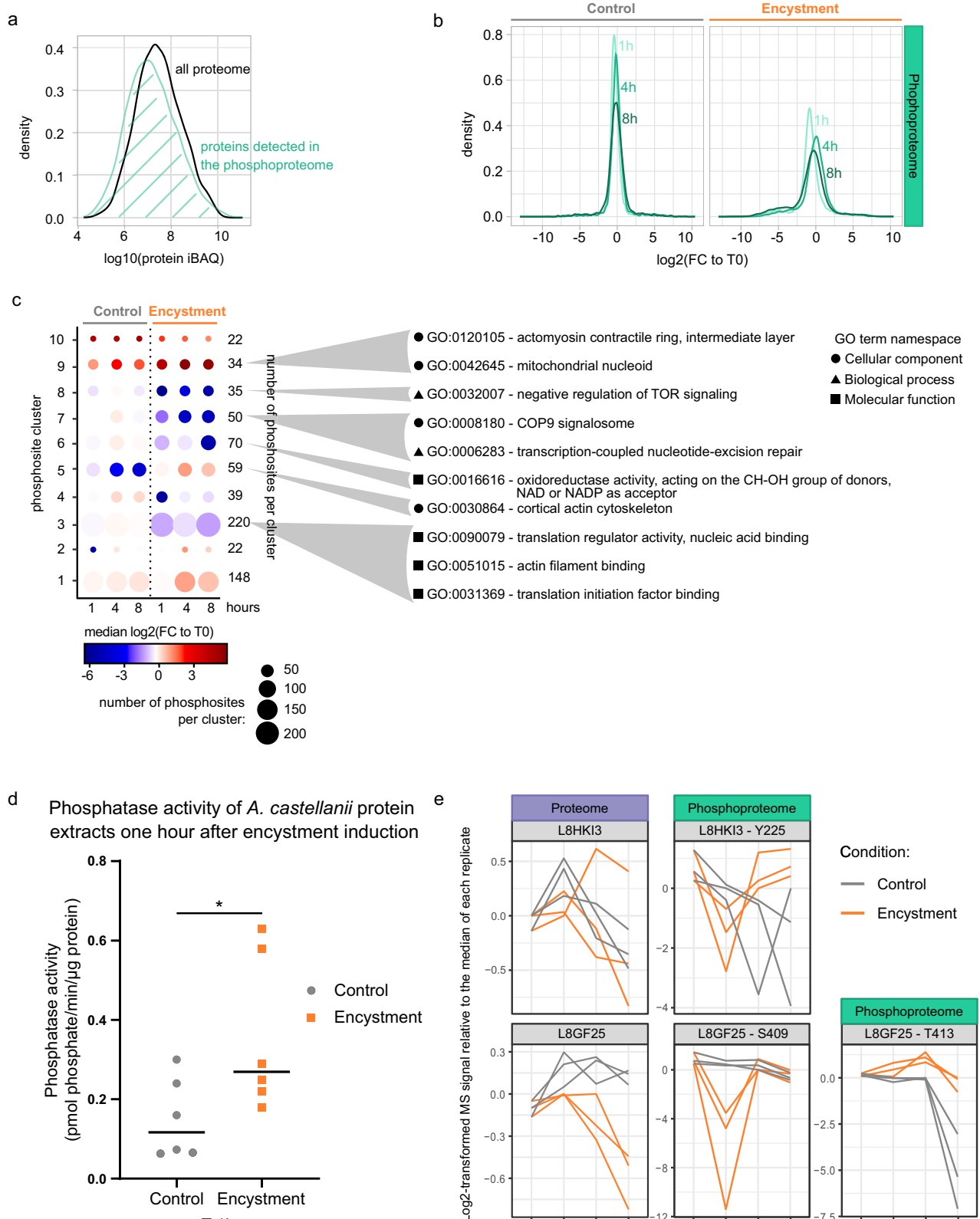

encystment, respectively. Mechanisms involving cytoskeleton rearrangement could be activated as soon as one hour after induction. Two GO terms belonging to the category "molecular function" and associated with translation were enriched in cluster 3: GO:0031369—translation initiation factor binding and GO:0090079— translation regulator activity, nucleic acid binding. This observation is in line with the one made with GO terms enrichment analysis performed on the transcriptomic and proteomic data (Figs. 2b and 3d). Moreover, the GO negative regulation of TOR signaling (GO:0032007) was enriched in cluster 8. This pathway is known to regulate cell growth/proliferation and autophagy, which is induced during the encystment of many FLAs[35,36].

**Fig. 4 Phosphoproteomics of *A. castellanii* encystment. a** Normalized distribution of the protein relative quantities (iBAQ) from the entire proteome and the subset of proteins also quantified in the phosphoproteome (in black and turquoise, respectively). **b** Normalized distribution of the log2-transformed fold changes relative to T0 control in the phosphoproteome in the control condition (left) and during encystment (right). Time points are indicated next to the corresponding density. **c** Clusters of significantly regulated phospho-peptides. The dot plot is color-coded with the mean of log2-transformed fold change relative to the control condition (0 h) for each cluster during encystment and in the control condition (1, 4, 8 h). The size of each dot is proportional to the number of phosphorylation sites per cluster (indicated on the right). Significantly enriched GO terms in the different protein clusters are present on the right side of the plot (cellular component, biological process and molecular function are represented with a circle, a triangle and a square, respectively). The detailed outputs of the enrichment can be found in Supplementary Data 2 file. **d** Dot plot representing the phosphatase activity (in pmol phosphate/min/μg) of *A. castellanii* protein samples obtained one hour after encystment induction ("E", orange) or growth medium renewal ("C", gray). $n = 6$ independent experiments; median values are presented by a horizontal bar; "*": Mann–Whitney one-sided test $p$-value = 0.0325. **e** Log2-transformed MS signal relative to the median of each replicate for the proteins discussed in the text and their regulated phosphorylation sites. Time points are indicated in the bottom: 0 h control and 1, 4, 8 h in control or encysting condition (gray and orange, respectively).

The phosphorylation status of specific sites is tightly regulated by the activities of kinases and phosphatases. In order to identify encystment-responsive phosphorylation motifs, we aligned the peptide sequences surrounding regulated phosphorylation sites and plotted the residues found on each position with a size proportional to their number of occurrences ($+/-15$ residues surrounding the site) using pLogo[32] (Supplementary Fig. 7). We did not observe any major phosphorylation motif surrounding phospho-threonines (Supplementary Fig. 7, right). Similarly, we did not identify any phospho-threonine-specific motif in up- or downregulated phosphorylation sites (Supplementary Fig. 7). A motif-x analysis[37] using the same data did not reveal any significantly enriched motif either, as a $p$-value threshold of $1 \times 10^{-6}$ would be required to maintain a low false-positive rate in such a motif search (Supplementary Data 6).

## Discussion

Encystment is a differentiation process that allows species sustainability by increasing microorganisms' resistance to environmental stresses. While encystment is widely spread across organisms, little is known about the molecular players involved. Difficulties to induce encystment in vitro and the limited number of available genomes, genetic and molecular tools for numerous encysting microorganisms are important barriers to decipher mechanisms leading to the formation of cysts. To overcome these limitations, we developed a proteogenomics approach that provides the first atlas of transcripts, proteins, and phosphoproteins differentially regulated during the earliest steps leading to encystment of *A. castellanii*.

We assembled a de novo transcriptome containing 166,782 transcripts, belonging to 55,723 genes. This number seems overestimated, as the reference genome and the recently published genome of Matthey-Doret et al.[38] only contain 14,974 and 15,497 genes, respectively. This high gene number could be the consequence of Trinity generation of isoforms and the presence of non-coding RNA. Sequence mapping can also lead to an overestimation of protein number due to protein isoforms and/or cleavages. However, our de novo approach was preferable as the BUSCO-completeness scores for the reference genome remain quite low. Based on the custom database for proteomic analysis our strategy allowed us to create using the newly built transcriptome, we identified 2701 protein groups absent from the reference proteome of *A. castellanii*, suggesting that the over-estimation may not be as extreme. Our dataset provides a unique resource of protein sequences that may be specifically expressed during encystment.

The global analysis of the three omics approaches showed that, in contrast to proteins, the quantities of transcripts and phosphorylation sites were modified as early as one hour after triggering encystment. Changes to the proteome were more gradual,

and mainly observable after 8 h. The temporal kinetic response to starvation used to induce encystment, especially at the transcriptomics level, allowed us to infer subsets of transcripts for which the expression was restricted to specific time points. From this, it seems that *A. castellanii* rapidly responds to starvation through the regulation of protein phosphorylation and gene expression at the level of transcription.

*A. castellanii* is predicted to encode the largest number of protein kinases among amoebozoans[5]. By providing the first phosphoproteome for this organism, we highlight complementary information about processes regulated during encystment and for which transcript and protein quantities do not change. Strikingly, we uncovered a specific increase in global phosphatase activity, one hour after starting starvation that was associated with a decrease of phosphorylated sites during encystment. Such regulation of the phosphatase activity was hinted at over 50 years ago by Griffiths et al., who found that the specific acid phosphatase activity increased by 34% in the first 3 h of encystment before returning to the baseline[39].

To identify the contribution of the different kinase and phosphatase families during encystment, it is important to distinguish protein regulation from specific enzymatically driven phosphorylation or dephosphorylation events. Today, specific sample preparations and MS-based strategies allow the calculation of phosphorylation occupancies in system-wide analysis. It can be based on protein/peptide labeling followed by phosphatase treatment[40,41], or more site-specific targeted MS[42]. These require high amounts of protein sample, and/or synthesis of specific phosphorylated peptides based on a priori knowledge of post-translational modifications expected in a sample. We believe that the RNA-seq-derived protein sequences, the proteome, and phosphoproteome presented in the present study is a unique resources for planning such studies.

We could hypothesize that the early regulation of transcription during encystment results from the repression of transcription factors through phosphorylation. Some of these factors, and other molecules that regulate gene expression, are differentially phosphorylated in our phosphoproteomic data. The glycogen synthase kinase 3 (GSK3; L8HKI3; gene ACA1_289050), differentially phosphorylated in position Y225 in our analysis (cluster 3) (Fig. 4e), is a widespread kinase with a broad array of known substrates, which regulates many transcription factors such as Fos/Jun AP-1 or p53[43]. Harwood et al. described that this enzyme was required in *D. discoideum* to allow the cAMP's promotion of pre-spore over stalk differentiation[44]. GSK3 could also affect other steps of *D. discoideum* development, as genes from the 2C family would require its presence for optimal expression[45]. More recently, Kawabe et al. reported that in *Polysphondylium pallidum*, a Dictyostelid from a group that conserved its ability to encyst, GSK3 promotes development into fruiting bodies over encystment[46]. Another protein of interest is a putative Myb-like

DNA binding protein, according to UniprotKB annotation (L8GFZ5; gene ACA1_314230) for which we observed in our tested condition both a downregulated (cluster 4, position S409) and an upregulated (cluster 5, T413) phosphorylation site (Fig. 4e). Such a protein has been reported to control the expression of many stage-specific genes involved in cyst formation in *E. histolytica*[47]. In *Toxoplasma gondii*, the Myb-like transcription factor BFD1 was shown to be necessary and sufficient to induce encystment of *T. gondii*[48]. In our condition, the level of the putative Myb-like DNA binding protein only decreased 4 h after induction of encystment enabling a possible induction of transcription of encysting genes at one hour.

Proteasome-related GO terms, ubiquitin-dependent, and independent proteasome-mediated protein catabolic processes were enriched in modules that were upregulated after encystment induction, providing insight into the regulation of protein turnover upon encystment. The importance of the proteasome and ubiquitin-conjugating genes was previously shown as lactacystin, a non-peptidic proteasome inhibitor, prevents encystment of *A. castellanii*[49]. In our transcriptome, ubiquitin-related transcripts were enriched in modules that showed upregulation 1, 4, and 8 h after encystment induction. Along with these data, we also observed the dephosphorylation (cluster 7) of S281 in a subunit of the COP9 signalosome (L8HEE3; gene ACA1_074760), which controls the cellular ubiquitylation status through the remodeling of cullin-RING ubiquitin ligases and its deubiquitylating activities[50]. The presence of the COP9 signalosome in *Acanthamoeba* was recently reported[51]. Our data suggest that the COP9 signalosome could regulate ubiquitylation during encystment. Furthermore, the putative role of the COP9 signalosome in encystment indicates the potential role of other post-translational modifications such as neddylation.

Autophagy is another degradative process that has been linked to encystment. Autophagy initiation can be regulated by a serine/threonine kinase TOR (Target of Rapamycin), which controls cell growth by promoting anabolic processes such as protein synthesis, transcription, and ribosome biogenesis. It also antagonizes catabolic processes including mRNA degradation and autophagy[52]. TOR activity is regulated by nutritional signals and environmental stresses[53]. In our phosphoproteomic data, the GO term "negative regulation of TOR signaling" was enriched in cluster 8. This cluster displayed a decrease of phosphosite abundance as early as one hour after encystment induction. This enrichment is due to the presence of four phosphosites present on L8HGH9, a retinoblastoma-associated protein A domain-containing protein; L8H4J3, a sestrin-like protein; L8GYH9, a putative ROCO family protein kinase; and L8GW34, a putative phosphatase. Interestingly, the sestrin-like protein from *D. discoideum* has already been described as involved in autophagy under starving condition[54]. The regulation of TOR signaling suggests that regulation of autophagy during encystment of *A. castellanii* occurs upstream of the regulation of the *atg* genes that have been shown to be essential for encystment[14].

Cyst wall formation is a mandatory process of encystment. This wall is composed of glycoproteins, such as lectins and carbohydrate molecules as cellulose. Consistent with the induction of cyst wall formation, we found that transcripts related to carbohydrate metabolic processes were upregulated as early as one hour after starvation. Cellulose synthase and xylose isomerase, which have also been shown to play a role in cyst wall formation, were upregulated at transcript level[13]. The protein turnover observed during encystment could provide material to synthesize the new cystic proteins. Lectins are an important part of cyst walls. Three sets of these proteins have been identified as the most abundant in *A. castellanii* cyst wall, namely the Jonah, Luke, and Leo families[19]. The assembly of these different lectins with

glycopolymers is a crucial step of *A. castellanii* cyst wall formation. Among the 31 lectins described by Magistrado et al.[19], 23 were likely present in our transcriptomic data, and 16 were significantly differentially expressed. The corresponding transcripts were distributed in modules 1, 2, 18, and 19. However, only six of these proteins were identified in our proteome: one Jonah (L8GRM4), four Luke (L8HBQ6, L8GV32, L8HC87, and L8HC82) and one Leo (L8H602). None of them were significantly regulated at the protein level in our experimental conditions. This can be due to timing since their peak of expression may occur later than eight hours after encystment induction[19].

In conclusion, we provide the relative quantities of transcripts, proteins, and phosphorylation sites during *A. castellanii* encystment, alongside their functional annotation and statistical analysis. This large-scale molecular atlas of *A. castellanii* encystment completes our knowledge of this organism during short-term encystment and provides a first glimpse at specific phosphoregulations occurring in this context. These data may be translated to the study of encystment in other organisms. Most of all, we believe that this resource can be leveraged to offer biocidal opportunities against *A. castellanii*.

## Methods

**Cell lines**. *Acanthamoeba castellanii* strain Neff (ATCC 30010™) were maintained in peptone yeast glucose (PYG) medium (2% proteose peptone, 0.1% yeast extract, 0.1 M glucose, 0.1% sodium citrate dihydrate, 0.4 mM CaCl₂, 4 mM MgSO₄, 0.05 mM Fe(NH₄)₂(SO₄)₂ 6H₂O, 2.5 mM K₂HPO₃, 2.5 mM NaH₂PO₃, pH 6.5), at 30 °C without shaking. All experiments described hereafter were performed with amoebae subcultured <20 times.

**Encystment kinetic using flow cytometry**. *A. castellanii* trophozoites were seeded into six-well microplates at a density of $2 \times 10^5$ cells in 3 mL of PYG medium and incubated at 30 °C overnight. Cells were cultivated under two different conditions for growth and encystment. For growth, PYG medium was replaced by 3 mL of fresh PYG medium. For encystment, cells were rinsed twice with 3 mL of Page's Amoeba Saline (PAS) buffer (0.1% sodium citrate dihydrate, 0.4 mM CaCl₂, 4 mM MgSO₄, 0.05 mM Fe(NH₄)₂(SO₄)₂ 6H₂O, 2.5 mM K₂HPO₃, 2.5 mM NaH₂PO₃, pH 6.5) and then 3 mL of encysting buffer (0.1 M KCl, 0.4 mM CaCl₂, 1 mM NaHCO₃, 8 mM MgSO₄, and 20 mM 2-amino-2-methyl-1,3-propanediol, pH 8.8) were added to wells. Cells were incubated at 30 °C for 24, 48, 72 or 168 h to evaluate *A. castellanii* encystment. After incubation, we always verified microscopically the presence of double-walled cysts before proceeding to the next steps. The supernatant of cells cultivated under the growth condition was removed and 3 mL of PBS containing EDTA (137 mM NaCl, 2.7 mM KCl, 10 mM NaHPO₄, 1.8 mM KH₂PO₄, 2 mM EDTA, pH 7.4) were added to each well. Cells under the encystment condition were treated with EDTA with a final concentration of 2 mM. Then, cells cultivated in both conditions were detached using cell scrapers, and suspensions were transferred to 5 mL, 75 × 12 mm, flow-cytometry tubes after filtration on 70 μm cell strainers. Cells were stained using 10 μL of calcofluor white (Becton, Dickinson and Company, USA), a dye that binds to 1,4-β-glucans such as cellulose, to distinguish trophozoites from cysts. Samples were analyzed on a Cytek® Aurora (Cytek Biosciences, USA) using the SpectroFlo® 2.2 software (Cytek Biosciences, USA). The flow rate was set to medium (30 μL/min) and the FSC and SSC gains were set to 20 and 40, respectively. Cyst fluorescence associated with calcofluor white was observed on the V4 channel (center wavelength: 473 nm; bandwidth: 15 nm; wavelength start: 466 nm; and wavelength end: 481 nm) and at least 10,000 events were recorded. The gating strategy used is detailed in Supplementary Fig. 8.

**RT-qPCR of encysting *A. castellanii***

*Primer design and efficiency*. Specific primer pairs were designed to target transcripts identified in this study using Primer-BLAST (NCBI) (Supplementary Data 7). Sequences from our transcriptome were used with the following parameters: PCR product size from 70 to 200 bp, primer melting temperatures ranging from 57 to 63 °C with an optimal melting temperature at 60 °C and no exon/intron selection.

Every primer pair were synthetized by Eurogentec (Belgium). Their efficiency was evaluated using *A. castellanii* genomic DNA. Briefly, a standard curve was constructed with qPCR results obtained with the designed primer pairs and a five-fold serial dilution of *A. castellanii* genomic DNA. The PCR amplification efficiency (E) was then determined for each primer pair using the slope of the standard curve. According to MIQE guidelines[55], all primer pairs designed with an efficiency under 80% or above 110% were discarded. Reactions for standard curve construction were performed using LightCycler® FastStart DNA Master plus SYBR Green I (Roche Applied Science, Germany) on a LightCycler® 480 Instrument II (Roche Applied Science). Reactions were prepared in a final volume of 10 μL

containing 5 μL of 2X Master plus SYBR Green I, 2 μL of PCR grade water, 2 μL of diluted gDNA and 0.5 μL of each primer (10 μM).

The following amplification protocol was used: an initial denaturation step at 95 °C for 5 min followed by a three-step thermal cycling profile consisting of denaturation at 95 °C for 10 s, primer annealing at 60 °C for 10 s and extension at 72 °C for 10 s. A total of 45 cycles were performed followed by a melting curve analysis, ranging from 65 to 95 °C, to verify the specificity of each primer pair.

*Sample preparation and RT-qPCR.* A. castellanii trophozoites were seeded into seven 25 cm² culture flasks at a density of $5 \times 10^5$ cells in 10 mL of PYG medium and incubated at 30 °C overnight. For growth conditions, PYG medium was replaced by 10 mL of fresh PYG medium. For encysting condition, cells were rinsed twice with 10 mL of PAS and 10 mL of encysting buffer were added to the culture flasks. Cells were incubated at 30 °C for 1, 4, or 8 h. Following incubation, cells were rinsed twice with PAS and total RNA was extracted with a RNeasy Mini Kit (QIAGEN, Germany). As a control condition ("T0"), after overnight incubation, cells were rinsed twice with 10 mL of PAS and total RNA extraction was carried out directly.

Total RNA was extracted with a RNeasy Mini Kit (QIAGEN, Germany). All RNA samples were treated with a RNase-free DNase I contained in the TURBO DNA-free™ kit (Invitrogen, USA) for further DNA decontamination. Reverse transcription was carried out using the GoScript™ Reverse Transcriptase kit (Promega, USA) following the manufacturer's recommendations and using 100 ng of RNA matrix, 5 mM of $MgCl_2$ and a 1/1 mix of Oligo(dT)15 and random primer. Obtained products were diluted to ten-fold and according to Fluidigm's protocol (quick reference PN 100-5875 B1). cDNA were then diluted five-fold with Tris-EDTA buffer. Gene expression was then assessed on 48 × 48-well integrated fluidic circuit plate (GE Dynamic Array) using a Biomark™ HD system (Fluidigm Corporation, USA)[56]. Fold changes of the studied transcripts were calculated by the ΔΔCt method using Fluidigm Real-Time PCR Analysis software (Fluidigm Corporation). The control sample used was P0/T0 and reference genes used were HPRT, G6PD, GAPDH, RasC, and 18S (SQV and ST4 primers) as described by Köhsler et al.[57]. A total of 25 genes on six series of seven samples were analyzed.

## Transcriptomic analysis

*A de novo transcriptome assembly.* A total of 21 samples (7 conditions in triplicate) were sequenced at the GENEWIZ genomics service company (Leipzig, Germany), using the paired-end Illumina HiSeq libraries (2 × 150 bp reads). Preparation of the RNA library included a double rRNA depletion with both the Ribo-Zero rRNA removal kit (Human/Mouse/Rat) and the Ribo-Zero rRNA removal kit (Bacteria). All raw data are available in the National Center for Biotechnology Information (NCBI) Sequence Read Archive (SRA) database under the BioProject database identifier PRJNA794325.

Data were processed using a FAIR workflow built specifically for this project. This pipeline and the assembled transcriptome are publicly accessible: https://github.com/ifremer-bioinformatics/FLORA. Briefly, raw RNA-Seq read quality was inspected using FASTQC (https://www.bioinformatics.babraham.ac.uk/projects/fastqc/). An error correction step was performed using Rcorrector (version 1.0.3)[58] with default parameters in order to correct sequencing errors in reads or to remove incorrect reads. To ensure to remove all rRNA contamination, all reads were mapped against the SILVA database (version 132)[59] using Bowtie2 (version 2.2.1)[60] with the '--very-sensitive' option. Only unmapped reads (corresponding to non-rRNA reads) were kept for further processes. Then, low-quality and short reads, sequencing adaptors, and "N" nucleotides were removed using TrimGalore (version 0.6.1) (https://www.bioinformatics.babraham.ac.uk/projects/trim_galore/). The Trinity package was used to assemble and annotate 1,075,406,542 reads (sum of paired reads) that represent ~70% of the initial reads.

A completeness analysis of the A. castellanii reference transcriptome available (GCA_000313135.1, BioProject PRJNA66753)[5] against the BUSCO Eukaryota_odb10 database (version 3.0.2)[61] was performed to define our assembly strategy. The obtained metrics were not satisfactory, thus we decided to perform a de novo assembly of our data to obtain a more recent and exhaustive transcriptome. This de novo assembly was carried out with Trinity (version 2.6.6)[62]. The presence of possible contamination in our assembly was further investigated with taxon-annotated GC content-coverage plot using Blobtools[63]. Taxonomic assignment was based on the taxonomic classification of their BLASTn and BLASTx results ensuring the most accurate results as recommended in the tool documentation. Finally, the transcriptome was validated by mapping back the preprocessed reads to the assembly using Bowtie2, when nearly 100% mapped properly.

*Differential expression analysis.* Transcript quantifications were estimated by mapping all pre-processed RNA-Seq paired reads for each sample to the final assembled transcriptome using Bowtie2. The expression levels of mapped paired reads were normalized using the TMM normalization method with RSEM[64]. No filtering was done based on the expression values. Quality check of replicates was performed using the post-transcriptome assembly downstream analyses tools of Trinity. The normalized reads were then used for the differential expression analyses using the DESeq2[26] at the transcript level using the Trinity 'run_DE_analysis.pl' script. Differentially expressed transcripts were identified using the Trinity

'analyze_diff_expr.pl' script with default parameters: an absolute log2 fold change ≥ 4 and a false discovery rate (FDR) p-value ≤ 0.001. Differentially expressed transcripts were further clustered using the WGCNA R package[27] that allows to group all transcript based on their expression pattern.

## Mass spectrometry-based analysis

*Protein sample preparation.* A. castellanii trophozoites were seeded into seven 150 cm² culture flasks at a density of $2 \times 10^6$ cells in 30 mL of PYG medium and incubated at 30 °C overnight. For growth conditions, the medium was replaced by a fresh PYG medium. For encysting condition, cells were rinsed twice with PAS and cultured within the encysting buffer. Cells were incubated at 30 °C for 1, 4 or 8 h, rinsed twice with PAS buffer and proteins were extracted using a urea-based extraction protocol. Briefly, cells were harvested using a cell-scrapper, placed in a 50 mL tube and centrifuged at $1000 \times g$ for 10 minutes. Pellets were dissolved in 1.5 mL of extraction buffer (Tris-HCl 50 mM, urea 8 M, pH8). Mechanical lysis was performed using a FastPrep-24™ 5 G system (MP Biomedicals, USA) with the following settings: 3 steps of 30 s at speed 6, with a 5 min incubation on ice between each cycle. A. castellanii lysates were then centrifuged at $16,000 \times g$ for 10 minutes. One mL of each supernatant was transferred to a new tube and stored at −80 °C, before further analysis. As control condition (T0), after overnight incubation, cells were rinsed twice with 10 mL of PAS, and protein extraction was carried out directly. Protein concentration was measured using a Qubit® protein assay kit with Qubit® 2.0 Fluorometer (Invitrogen, USA).

*In-solution protein digestion.* One milligram of protein extract was solubilized in 8 M urea, 100 mM Tris HCl pH 8.5, then disulfide bonds were reduced with 5 mM Tris(2-carboxyethyl) phosphine (TCEP) for 30 min and alkylated with 10 mM iodoacetamide for 30 min at room temperature (RT) in the dark. Protein samples were then incubated with rLys-C Mass Spec Grade (Promega, Madison, WI, USA) for 5 h at 30 °C for the first digestion (ratio 1:50). Then samples were diluted below 2 M urea with 100 mM Tris HCl pH 8.5 and Sequencing Grade Modified Trypsin (Promega, Madison, WI, USA) was added for the second digestion overnight at 37 °C (ratio 1:100). Digestion was stopped by adding 5% of formic acid (FA) and peptides were desalted and concentrated on Sep-Pak light $tC_{18}$ SPE cartridge (Waters, Milford, MA, USA) according to manufacturer's instructions. Peptides were eluted using 50% acetonitrile (ACN), 0.1% FA. Purified peptides were lyophilized and kept at −80 °C until further used.

*Phosphopeptide enrichment.* TiO2 phospho-enrichment was performed using a modified version of Matheron et al. protocol[33,34].

A slurry of 10 mg/mL of $TiO_2$ beads (Sachtopore-NP TiO2, 5 μm, 300 Å, Sachtleben, Duisburg, Germany) was freshly prepared and kept in 30% ACN,0.1% trifluoroacetic acid (TFA) until used.

One milligram of dried tryptic peptides was re-suspended in $TiO_2$ loading buffer: 80% ACN, 6% TFA, 40 mg/mL glycolic acid. The sample was sonicated to ensure a higher yield of resuspension.

A 10:1 ratio (beads/peptides) was used for enrichment. Suitable $TiO_2$ beads volume was equilibrated in loading buffer for 5 min at RT. Samples were mixed with beads and briefly sonicated. Incubation was performed under gentle agitation on a rocking device for 30 min at RT. After 1 min of centrifugation, the supernatant was discarded. Beads were first washed in 80% ACN, 6% TFA, gently vortexed, and centrifuged for 1 min (discarded supernatant). Beads were washed a second time with 50% ACN, 0.1% TFA, gently vortex, and centrifuged (discarded supernatant). Phosphopeptides were eluted from $TiO_2$ beads using 10% $NH_4OH$ solution into a new tube containing 20% FA. Eluate fractions were lyophilized and then $C_{18}$ stage-tipped[65] for clean up until mass spectrometry analysis.

*Mass spectrometry analysis.* Tryptic peptides were analyzed on a Q Exactive Plus mass spectrometer (Thermo Scientific, Bremen) coupled with an EASY-nLC 1200 chromatography system (Thermo Scientific, Bremen). The sample was loaded and separated at 250 nL min⁻¹ flow rate on a home-made packed 50 cm nano-HPLC column (75 μm inner diameter) with $C_{18}$ resin (1.9 μm particles, 100 Å pore size, Reprosil-Pur Basic $C_{18}$-HD resin, Dr. Maisch GmbH, Ammerbuch-Entringen, Germany) and equilibrated in 97% solvent A ($H_2O$, 0.1% FA) and 3% solvent B (ACN, 0.1% FA).

Input sample peptides (proteome) were eluted with a linear gradient from 3 to 22% buffer B in 160 min. This was followed by a stepwise increase of buffer B to 50% B in 70 min and finally to 90% in 5 min.

Enriched phosphopeptides were eluted with a linear gradient from 3 to 13% buffer B in 40 min. This was followed by a stepwise increase of buffer B to 37% B in 110 min and finally to 75% in 20 min.

Mass spectra were acquired with a Top10 data-dependent MS/MS acquisition mode with the XCalibur 2.2 software (Thermo Fisher Scientific, Bremen). Target values for the full-scan MS spectra were $3 \times 10^6$ charges in the *m/z* range 200–2000 with a resolution of 70,000 at *m/z* 200. Fragmentation of precursor ions was performed by Higher-energy C-trap Dissociation (HCD) with a normalized collision energy of 27 eV. MS/MS scans were performed at a resolution of 17,500 at *m/z* 200 with a target value of $1 \times 10^6$ and a maximum injection time of 20 ms. Charge state screening was enabled, and precursors with unknown charge state or a charge state of 1 and >7 were excluded. Dynamic exclusion was set to 35 s (Input

sample) or 40 s (Phosphopeptides) to avoid repeated sequencing of identical peptides.

*Data Processing for protein identification and quantification.* All data were searched using Andromeda[66] with MaxQuant software (version 1.5.3.8)[67,68] against a custom-generated database constituted of the non-redundant protein sequences from the Uniprot *A. castellanii* strain Neff proteome and the Trinity database (129,654 entries after curation), known mass spectrometry contaminants (MaxQuant contaminant database with the proteome of *Parachlamydia acanthamoebae* strain UV7 to control for contamination because this strain was used in the laboratory) and reversed sequences of all entries. The custom FASTA can be downloaded from zenodo.org (DOI 10.5281/zenodo.5869928). Andromeda searches were performed choosing trypsin as a specific enzyme with a maximum number of two missed cleavages. Possible modifications included carbamidomethylation (Cys, fixed), phosphorylation (Ser/Thr/Tyr, variable), oxidation (Met, variable) and Nter acetylation (variable). The mass tolerance in MS was set to 20 ppm for the first search then 6 ppm for the main search and 10 ppm for the MS/MS. The maximum peptide charge was set to seven. Five amino acids were required as minimum peptide length. The "match between runs" feature was applied for samples having the same experimental condition with a maximal retention time window of 0.7 min. One unique peptide to the protein group was required for the protein identification. A false discovery rate (FDR) cutoff of 1% was applied at the peptide and protein levels. The MaxLFQ, Maxquant's label-free quantification, algorithm was used to calculate protein intensity profiles across samples[69]. iBAQ feature was selected for further quantitative analysis. The MS proteomics data have been deposited to the ProteomeXchange Consortium via the PRIDE[70] partner repository with the project accession number PXD031788.

**Phosphatase activity.** The phosphatase activity in *A. castellanii* was quantified following the Serine/Threonine Phosphatase Assay System (Promega) protocol. Briefly, amoebae were incubated in PYG growth medium or in the encysting medium for 1 h. Then, proteins were extracted using a Tris-HCl Buffer (10 mM Tris, 0.05% β-mercaptoethanol, 1 mM EDTA, 1 mM protease inhibitors (Halt Protease inhibitor cocktail, Thermo Scientific), 0.05% Triton X-100, pH 7.4). The homogenate was centrifuged at 4 °C, 16,000 × g, for 1 h. The cell lysate was added to the Sephadex G-25 spin column and centrifuged at 4 °C, 600 × g, for 5 min. Ten µg of proteins were used to evaluate the phosphatase activity within a mixture of protein phosphatase-2A buffer (250 mM imidazole, 1 mM EDTA, 0.05% β-mercaptoethanol, 0.5 mg/mL of BSA, pH 7.4) and 50 µM of phosphopeptide (RRA(pT)VA). The reaction mixture was incubated at room temperature (RT) for 30 min. The reaction was stopped after the addition of molybdate dye/additive mixture at RT for 30 min. The absorbance of the samples was read at 630 nm using an Infinite M Plex Microplate Reader (Tecan, Switzerland). The statistical significance of the results was checked with a one-tailed Mann–Whitney test.

**Computational analysis.** The computational analysis of the proteome, the phosphoproteome, and their integration with the transcriptome were done with R v4.0.3 (2020-10-10) on x86_64-apple-darwin17.0 (64-bit) running under macOS Big Sur 10.16. The corresponding scripts are available on Zenodo with the DOI 10.5281/zenodo.5869928 under the license BSD 2-Clause "Simplified".

*Functional annotation.* Transcript and protein functional annotations were retrieved from eggNOG-mapper v2.0 (http://eggnog-mapper.embl.de) against the eggNOG v5.0 database with the default parameters using as input a protein FASTA file containing all protein sequences corresponding to the transcriptome or the refined non-redundant FASTA file utilized for MS search[30,31]. For each gene in the transcriptome, we kept the annotation corresponding to the best scoring ortholog across all transcripts (Trinity IDs). GO term enrichments were performed using a hypergeometric test with the full transcriptome or the full proteome as background for modules or proteome/phospho enrichment, respectively (go-enrich function of the R package GOfuncR v1.10.0 with n_randsets=100). Genes from significant child categories were removed with a FWER threshold of 0.1 (function refine).

*Statistical analysis of the proteome.* We used the MaxQuant "ProteinGroups.txt" table, removed proteins with <1 unique peptide or identified only with modified peptides, reverse hits, and potential contaminants. LFQ intensities were log2-transformed and realigned using median subtraction. Missing values were replaced only for the protein groups that had no missing values in one condition in a minimum of two replicates. Values entirely absent in one experimental condition (treatment + time point) (under detection threshold) were replaced using the QRILC method (R package imputeLCMD v2.0 with tune.sigma = 1); values absent of one replicate only (missing at random) were replaced using the slsa method (R package imp4p v0.9). For each protein group, we performed a two-sided paired t-test for each time point and an ANOVA test across all time points. p-values were corrected for multiple testing using the adaptative procedure of Storey. The significantly regulated proteins (q-value of the ANOVA and one of the two-sided paired t-test with a q-value ≤ 0.05) were clustered using hierarchical clustering (hclust function with method = "ward.D" applied on log2-transformed fold

changes relative to T0, tree cut with k = 10). GO term enrichment per cluster was performed as for transcript modules (see paragraph "functional annotation").

*Statistical analysis of the phosphoproteome.* We used the MaxQuant "Phospho (STY)Sites.txt" table. Phosphorylation sites on potential contaminants were removed as well as reverse hits, and we kept sites with a PEP ≤ 0.01 and a localization probability ≥ 0.75. Log2-transformed intensities were normalized per condition (treatment + time point) to conserve potential shift of phosphopeptide median signal. Missing value replacement and statistical analysis were performed as described for the proteome. In addition, we performed ANOVA contrast analyses to identify phosphorylation sites with time courses significantly different from the protein these belong to.

*Combined analysis of the three data sets.* Each majority protein ID (keeping only the accessions with the highest number of unique peptides) was matched to the corresponding transcript with the highest sum of signal across all samples. We confirmed that using the sum of signal over all transcripts instead gave similar results (Supplementary Fig. 4a, b).

**Reporting summary.** Further information on research design is available in the Nature Research Reporting Summary linked to this article.

## Data availability

All transcriptomic raw data generated in this study have been deposited in the NCBI SRA database, under the BioProject database identifier PRJNA794325. The MS proteomics data generated in this study have been deposited to the ProteomeXchange Consortium via the PRIDE partner repository, under the project accession number PXD031788. FASTA used for the computational analysis and the protein identification and quantification used in this study are available on Zenodo at https://doi.org/10.5281/zenodo.5869928. The raw data for all figures generated in this study are included in the supplementary data files or in the source data file. The following genomes were used in this study: *Acanthamoeba castellanii* strain Neff [GCA_000313135.1, BioProject PRJNA66753], *Acanthamoeba castellanii* strain Namur [GCA_903821525.1], *Acanthamoeba polyphaga* strain Linc Ap-1 [GCA_001567625.1], *Dictyostelium discoideum* [GCA_000004695.1], *Naegleria gruberi* [GCA_000004985.1]. Source data are provided with this paper.

## Code availability

All transcriptomic raw data generated in the study were processed using a FAIR workflow built specifically for this project and publicly accessible: https://github.com/ifremer-bioinformatics/FLORA. Scripts used for the computational analysis and the protein identification and quantification are available on Zenodo at https://doi.org/10.5281/zenodo.5869928.

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

## Acknowledgements

This work was supported by the Agence Nationale de la Recherche (grant ANR-17-CE13–00001–01 "Amocyst") (A.S-L.) and by the Novo Nordisk Foundation (grant NNF14CC0001) (L.J.J. and M.L-.P). A.S-.L. received a JEDI (justice, equity, diversity, and inclusion) award from Life Science Editors and we would like to thank Marie Bao for editorial assistance. We acknowledge the Ebioinf bioinformatics core facility of EBI for providing the computing infrastructure and resources. Finally, we thank Gwenaelle Quere and Amélie Jessu for technical support, Vincent Delafont for technical assistance and for the critical reading, and Lesley-Ann Giddings for proofreading the article.

## Author contributions

A.S.-L. conceived, designed, and coordinated the study. C.B., M.D., and A.S.-L. performed the in vitro experiments. M.L.-P., C.N., Q.G.G., B.M., and M.M. performed the in silico analysis. C.B., M.L.-P., C.N., M.D., Q.G.G., B.M., T.R., Y.H., L.J.J., M.M., and A.S.-L. contributed to the interpretation of results. C.B., M.L.-P., C.N., M.D., and A.S.-L. wrote the manuscript with input from all authors. C.B., M.L.-P., C.N., M.D., Q.G.G., B.M., T.R., Y.H., L.J.J., M.M., and A.S.-L. approved the manuscript.

## Competing interests

The authors declare no competing interests.
