## [Peer Review File · Nature Communications]

A time-resolved multi-omics atlas of *Acanthamoeba castellanii*
encystmentREVIEWER COMMENTS

Reviewer #1 (Remarks to the Author):

The authors use RNAseq, proteomics, and phosphoproteomics to characterize the early stages of encystation by *Acanthamoeba castellanii*, a free-living protist that causes keratitis and amoebic encephalitis. Their methods are vigorous, and their presentation is clear and does not over-interpret their data.

Their major findings include:

- 1) identification of many many more protein-encoding genes than previously annotated by the initial genome project.
- 2) demonstration of 18 patterns of gene expression during the transition from trophozoite to cyst, which suggests gene expression is coregulated in predictable and unpredictable ways.
- 3) demonstration that changes in gene expression and protein phosphorylation are fast, while changes in protein abundance is slow.
- 4) a series of stories, mostly included in the Discussion, suggesting transcription factors, kinases/phosphatases, actin-associated proteins, proteosomes, autosomes, and cell wall proteins/enzymes that are important for the encystation process.

This is an enormous amount of work that will be useful for all those investigating this poorly characterized protist. For example, the fasta file of proteins deposited in Zenodo makes it possible to correct protein predictions for AmoebaDB that are wrong, because of incorrect sequence assemblies or incorrect exon prediction.

Limitations to this work are:

- 1) 8 hour time point is the beginning of encystation that is not completed until 72 hours, so that many aspects of encystation are missed.
- 2) it would be great if data was placed in AmoebaDB, where it is much easier to access individual genes.

John Samuelson

Reviewer #2 (Remarks to the Author):

With this manuscript, the authors provide the first multi-omics atlas of *Acanthamoeba* encystment. The study was well-designed and offers a sound and valuable basis not only for all molecular studies on *Acanthamoeba* but also for studying encystment in other protists. The manuscript is very well written in most parts and is overall clear and easy to follow. To my opinion, it is suitable for publication after minor revision.

Specific comments:

While most parts of the manuscript are very well written, the first part of the introduction is, to my opinion, somehow clumsy and unnecessarily exaggerating the medical relevance of *Acanthamoeba*.

lines 38-39: firstly, not all FLA belong to the Amoebozoa, and secondly, the FLA are surely not the most important human pathogens. Here, *Entamoeba histolytica* has to be mentioned.

lines 41-42: this is too anthropomorphic, they do not prevent their eradication, in fact, FLA normally do not have contact to the human immune system or biocides; all this is purely accidental

lines 43-51: I suggest to focus here on *Acanthamoeba* or at least on the Amoebozoa;

Naegleria is not related at all and also its cysts are very different.

lines 52-59: I suggest to rephrase, to make it less dramatic and easier to understand. The reason why they may function as training grounds mainly is that they are phagocytotic eukaryotic cells and thus similar to human immune cells, e.g. macrophages.

lines 87-88: correct English; this would mean that the genome is easy to encyst.

line 90: analyses

lines 110-116: in the literature there have been several reports on Acanthamoeba producing immature cysts and so-called pseudocysts: has this been taken into account? I acknowledge that the authors used calcofluor staining, which likely binds best to mature cysts, but was the cyst wall checked microscopically (if doubled)?

line 194: processes

line 264: how were these time points actually chosen? What was the rationale for choosing exactly these time points?

line 272: cells

line 326: delete "be"

line 349: microorganisms? or unicellular organisms

line 379: amoebozoans

line 483: The Neff strain is known to not encyst easily (Parasitol Res (2008) 102:1069–1072), were the cultures pre-treated/subcultured somehow? This would be important to know for the reader.

lines 489-534: while most parts of the manuscript are very well written, this part needs revision of English language, numerous grammar mistakes.

line 550: the 2 should be subscripted

line 612: delete "of"

line 638: Peptides

line 653: vortexed and centrifuged

Reviewer #3 (Remarks to the Author):

The molecular mechanism of cyst formation is unclear yet in Acanthamoeba. The authors tried very vast effort to discover this secret. Up to date, most researchers studied on similar approach on Acanthamoeba faced a big problem that large portion of Acanthamoeba genes annotated as hypothetical protein. Even they showed high frequency during encystation, it is hard to investigate further function of the gene. In this paper, authors didn't mention about hypothetical protein. Please explain that how

much portin of genes annotated as hypothetical protein, and by how you overcome this problem to do further research.

Point-by-point response to the reviewers' comments

Reviewer #1 (Remarks to the Author):

The authors use RNAseq, proteomics, and phosphoproteomics to characterize the early stages of encystation by *Acanthamoeba castellanii*, a free-living protist that causes keratitis and amoebic encephalitis. Their methods are vigorous, and their presentation is clear and does not over-interpret their data.

Their major findings include:

- 1) identification of many many more protein-encoding genes than previously annotated by the initial genome project.
- 2) demonstration of 18 patterns of gene expression during the transition from trophozoite to cyst, which suggests gene expression is coregulated in predictable and unpredictable ways.
- 3) demonstration that changes in gene expression and protein phosphorylation are fast, while changes in protein abundance is slow.
- 4) a series of stories, mostly included in the Discussion, suggesting transcription factors, kinases/phosphatases, actin-associated proteins, proteosomes, autosomes, and cell wall proteins/enzymes that are important for the encystation process.

This is an enormous amount of work that will be useful for all those investigating this poorly characterized protist. For example, the fasta file of proteins deposited in Zenodo makes it possible to correct protein predictions for AmoebaDB that are wrong, because of incorrect sequence assemblies or incorrect exon prediction.

We thank the reviewer for his thorough evaluation of our manuscript and for his positive comments.

Limitations to this work are:

- 1) 8 hour time point is the beginning of encystation that is not completed until 72 hours, so that many aspects of encystation are missed.

We agree with the comment of reviewer 1 that encystation is generally not completed until 72h. However, as stated lines 21-22 or 100-103 of the manuscript, we chose to focus on the early steps of encystment to study molecular regulations that underly –and may be responsible for—*A. castellanii* encystment.

We inserted the following sentences to explain the choice of our time-points:

“We considered the 1h time-point because it precedes the maximum expression of the trehalose synthase gene observed at 2h in encysting cells²⁴. We opted for 4h and 8h, as Hirukawa *et al.* did not detect the cyst-specific protein CSP21, a marker of *A. castellanii* encystment, at these time-points²⁵.” (lines 123 -126)

2) it would be great if data was placed in AmoebaDB, where it is much easier to access individual genes.

We thank the reviewer for this suggestion. Transcriptome and proteome were submitted to AmoebaDB. We are still in the process of making them available to the community, in the meantime they are listed in the list of data sets that the AmoebaDB team is in the process of loading (<https://amoebadb.org/amoeba/app/static-content/dataInprogress.html> – see screenshot below).

Dataset Summary	GCA number	Citation	Short Attributi...	Data Type	Source DB Link	Release #
17. RNA seq transcriptome of encystation in A. castellanii Neff at 0, 1, 4, and 8 hours in encysting conditions with controls. All samples analyzed in triplicate.			Bernard C. et al. 2022	RNA-seq	https://www.ncbi.nlm.nih.gov/bioproject/PRJNA794325	Build 60
18. A. castellanii Neff proteome during encystation			Bernard C. et al. 2022	Quantitative proteomics		Build 60

John Samuelson

Reviewer #2 (Remarks to the Author):

With this manuscript, the authors provide the first multi-omics atlas of Acanthamoeba encystment. The study was well-designed and offers a sound and valuable basis not only for all molecular studies on Acanthamoeba but also for studying encystment in other protists. The manuscript is very well written in most parts and is overall clear and easy to follow. To my opinion, it is suitable for publication after minor revision.

We thank the reviewer for this positive comment.

Specific comments:

While most parts of the manuscript are very well written, the first part of the introduction is, to my opinion, somehow clumsy and unnecessarily exaggerating the medical relevance of Acanthamoeba.

We have modified this part to improve readability (see below).

lines 38-39: firstly, not all FLA belong to the Amoebozoa, and secondly, the FLA are surely not the most important human pathogens. Here, Entamoeba histolytica has to be mentioned.

Thank you. We have reworded these lines (37 to 40) to state: "Among protist, some free-living and parasitic amoebae, such as *Acanthamoeba castellanii* and *Entamoeba histolytica*, can undergo encystment".

lines 41-42: this is too anthropomorphic, they do not prevent their eradication, in fact, FLA normally do not have contact to the human immune system or biocides; all this is purely accidental

Thank you for raising this point. We have made the sentence clearer to the following: "This ensures species sustainability by allowing them to survive in the environment due to cysts high resilience" (lines 40 - 43).

lines 43-51: I suggest to focus here on Acanthamoeba or at least on the Amoebozoa; Naegleria is not related at all and also its cysts are very different.

Thank you for this suggestion. We have modified the manuscript accordingly to state: "For example, the genus *Acanthamoeba* was reported to cause fatal amoebic meningoencephalitis" (lines 47 - 48).

lines 52-59: I suggest to rephrase, to make it less dramatic and easier to understand. The reason why they may function as training grounds mainly is that they are phagocytotic eukaryotic cells and thus similar to human immune cells, e.g. macrophages.

This part has been shortened as follows: "In addition, some bacteria have been shown to become more virulent and more resistant to biocides and antibiotics after their passage through amoebae." (lines 57- 58).

lines 87-88: correct English; this would mean that the genome is easys to encyst.

The sentence has been reformulated to correct this error: "For these reasons, we investigated in this study the encystment of *A. castellanii*, a model eukaryote for cellular biology, as it readily undergoes encystment in starvation conditions and has an annotated genome available." (lines 88-90)

line 90: analyses

This error has been corrected (line 92).

lines 110-116: in the literature there have been several reports on Acanthamoeba producing immature cysts and so-called pseudocysts: has this been taken into account? I acknowledge that the authors used calcofluor staining, which likely binds best to mature cysts, but was the cyst wall checked microscopically (if doubled)?

Yes, Acanthamoeba does produce pseudocysts, and this is something we have taken into account. We always checked microscopically the state of the cysts before harvesting cells for further experiments, and we always observed the presence of cyst walls composed of two layers. This information has been included in the manuscript (lines 499 to 501).

line 194: processes

This error has been corrected (line 197).

line 264: how were these time points actually chosen? What was the rationale for choosing exactly these time points?

Thank you for this comment. To clarify, we have inserted the following sentences:

"We considered the 1h time-point because it precedes the maximum expression of the trehalose synthase gene observed at 2h in encysting cells²⁴. We opted for 4h and 8h, as Hirukawa *et al.* did not detect the cyst-specific protein CSP21, a marker of *A. castellanii* encystment, at these time-points²⁵." (lines 123 -126).

line 272: cells

This error has been corrected (line 275).

line 326: delete "be"

This error has been corrected (line 329).

line 349: microorganisms? or unicellular organisms

We corrected into microorganisms (line 354).

line 379: amoebozoans

This error has been corrected (Line 381).

line 483: The Neff strain is known to not encyst easily (Parasitol Res (2008) 102:1069–1072), were the cultures pre-treated/subcultured somehow? This would be important to know for the reader.

As shown on the Supplementary Fig. 1c, we agree with the reviewer that the Neff strain encysts slowly. No special pre-treatment nor subculture were used in addition to the protocol described (lines 489-515). We have generated several frozen stocks of *A. castellanii*. In that way, we never use amoebae subcultured for more than 20 times. Once they reach that point, we start from a new stock. This information has been included in the manuscript (lines 486-487).

lines 489-534: while most parts of the manuscript are very well written, this parts needs revision of English language, numerous grammar mistakes.

This part of the manuscript has been corrected (lines 489 to 540).

line 550: the 2 should be subscripted

This error has been corrected (line 555).

line 612: delete "of"

This error has been corrected (line 620).

line 638: Peptides

This error has been corrected (line 644).

line 653: vortexed and centrifuged

These errors have been corrected (line 659).

Reviewer #3 (Remarks to the Author):

The molecular mechanism of cyst formation is unclear yet in *Acanthamoeba*. The authors tried very vast effort to discover this secret. Up to date, most researchers studied on similar approach on *Acanthamoeba* faced a big problem that large portion of *Acanthamoeba* genes annotated as hypothetical protein. Even they showed high frequency during encystation, it is hard to investigate further function of the gene. In this paper, authors didn't mention about hypothetical protein. Please explain that how much portion of genes annotated as hypothetical protein, and by how you overcome this problem to do further research.

Reviewer #3 raises two important points regarding organisms with poorly annotated genomes such as *A. castellanii*:

- genes annotated as hypothetical proteins are genes that are annotated from sequence similarity, but we do not have any evidence of their transcription/translation to functional proteins. In the Uniprot/TrEMBL proteome of *A. castellanii* (5th of May 2022), we found less than 20 proteins with evidence at protein level (field "Protein existence"), although this number is an underestimation because this species is not well annotated. Therefore, there are many hypothetical proteins. Our MS-based proteogenomics data set provides evidence of the presence of many proteins in *A. castellanii* during encysting conditions, therefore we focused our discussion on data at the protein level.

- genes with no functional annotation: many genes of *A. castellanii* do not have a known function. We retrieved functional annotation using eggNOG-mapper (<http://eggno-mapper.embl.de/>) that transfers functional annotations from orthologs. This allowed the annotation of ~1900 protein sequences that were not in Uniprot/TrEMBL (5000+ proteins). All these annotations are available in the supplementary tables of the manuscript, and we believe that these are helpful when mining our data. To clarify this point in the manuscript, we have modified lines 212 - 215 as follows: "With this custom database we identified 8,577 protein groups (proteins that cannot be distinguished by MS because identified by a common set of peptides) (Fig. 3a) among which 2,342 had no functional annotation retrieved using eggNOG-mapper."

Of course, hypothetical genes with unknown functions remain a crucial issue when studying organisms such as amoebas. Indeed, these are particularly difficult to engineer *in vitro*, and high-throughput screening through genetic manipulation remains out of the question. In our group, we are currently working on the silencing of a few targets that proved to be very challenging. In this context, generation of data sets such as the one we provide here help in reducing the number of uncharacterized genes to focus on by providing a list of genes that are expressed and regulated in a given biological context. Furthermore, knowing this biological context (here encystment), can aid in the design of future experiments for functional validation of these genes.

REVIEWERS' COMMENTS

Reviewer #2 (Remarks to the Author):

The manuscript has been improved significantly, I only have some very minor comments, particularly for the introduction, which still reads slightly clumsy:

line 19: I suggest to delete "that are pathogenic", this is unclear, I suggest to add this information later on, as specified below

line 21: I suggest to here include: ...in the facultatively pathogenic amoeba...

line 38: the plural is: amoebae (or amoebas; or amebas); I suggest to change to: Among protists, most free-living and parasitic amoebae,....

line 40: ...due to the high resilience of the cysts.

lines 47-48: this is unclear, in fact, only very rarely, the contamination comes from the cleaning solutions. Also blindness is a very rare result of AK. I suggest to delete this sentence and instead write something like: This disease typically occurs in contact lens wearers, linked to poor contact lens hygiene, but also after trauma and contact to contaminated water.

Point-by-point response to the reviewers' comments

Reviewer #2 (Remarks to the Author):

The manuscript has been improved significantly, I only have some very minor comments, particularly for the introduction, which still reads slightly clumsy:

We thank the reviewer for her/his evaluation, and we modified the manuscript according to his comments.

line 19: I suggest to delete "that are pathogenic", this is unclear, I suggest to add this information later on, as specified below

We deleted the specified part of the sentence (line 19).

line 21: I suggest to here include: ...in the facultatively pathogenic amoeba...

We included the reviewer's suggestion in the specified sentence (line 21).

line 38: the plural is: amoebae (or amoebas; or amebas); I suggest to change to: Among protists, most free-living and parasitic amoebae,....

This error has been corrected (line 38).

line 40: ...due to the high resilience of the cysts.

The sentence was modified according to the reviewer's suggestion (line 40).

lines 47-48: this is unclear, in fact, only very rarely, the contamination comes from the cleaning solutions. Also blindness is a very rare result of AK. I suggest to delete this sentence and instead write something like: This disease typically occurs in contact lens wearers, linked to poor contact lens hygiene, but also after trauma and contact to contaminated water.

We agree with the reviewer's comment and we replaced our sentence with his suggestion (line 48-49).